# Sex as a Predictor of Response to Immunotherapy in Advanced Cutaneous Squamous Cell Carcinoma

**DOI:** 10.3390/cancers15205026

**Published:** 2023-10-17

**Authors:** Nicholas Yeo, Benjamin Genenger, Morteza Aghmesheh, Amarinder Thind, Sarbar Napaki, Jay Perry, Bruce Ashford, Marie Ranson, Daniel Brungs

**Affiliations:** 1Illawarra Shoalhaven Local Health District (ISLHD), NSW Health, Wollongong, NSW 2500, Australia; 2Molecular Horizons, University of Wollongong, Wollongong, NSW 2500, Australia; 3School of Chemistry and Molecular Bioscience, University of Wollongong, Wollongong, NSW 2500, Australia; 4Nelune Comprehensive Cancer Centre, Randwick, NSW 2031, Australia; 5Graduate School of Medicine, University of Wollongong, Wollongong, NSW 2500, Australia; 6Anatomical Pathology, Wollongong Hospital, Wollongong, NSW 2500, Australia; 7Southern IML/Sonic Healthcare, Wollongong, NSW 2500, Australia

**Keywords:** cutaneous squamous cell cancer, immunotherapy, sex, t-cell, biomarkers

## Abstract

**Simple Summary:**

Cutaneous squamous cell carcinomas (CSCC) are the second most common skin cancer amongst Caucasians, accounting for up to 20–25% of skin cancers. Whilst a majority of CSCCs can be cured with surgery alone, approximately 3–5% of patients develop advanced CSCC, which encompass locally advanced tumours or tumours with distant metastatic spread. As CSCCs are highly immunogenic, there is a strong rationale for treatment with immunotherapy. Several phase II clinical trials have demonstrated the benefit of immunotherapy in patients with advanced CSCC. However, only half of patients with advanced CSCC respond to immunotherapy, and thus there is a need to identify predictors of response. In this study, we demonstrated inferior clinical outcomes in female patients with advanced CSCC treated with immunotherapy compared to their male counterparts. This clinical finding is supported with translational assays on pre-treatment biopsies, demonstrating the presence of fewer anti-tumour immune cells in the tumours of female patients.

**Abstract:**

Approximately 3–5% of patients with cutaneous squamous cell carcinoma (CSCC) develop advanced disease, accounting for roughly 1% of all cancer deaths in Australia. Immunotherapy has demonstrated significant clinical benefit in advanced CSCC in several key phase II studies; however, there are limited data for patients treated outside of clinical trials. This is particularly relevant in advanced CSCC, which is most often seen in elderly patients with significant comorbidities. Thus, we aim to describe our experience with immunotherapy in a cohort of patients with advanced CSCC in Australia. We retrospectively reviewed all advanced CSCC patients treated with immunotherapy within the Illawarra and Shoalhaven Local Health District. Among the 51 patients treated with immunotherapy, there was an objective response rate (ORR) of 53% and disease control rate (DCR) of 67%. Our most significant predictor of response was sex, with male patients more likely to have better responses compared to female patients (DCR 85% vs. 41%, *p* < 0.0001), as well as improved progression-free survival (HR 4.6, 95%CI 1.9–10.8, *p* = 0.0007) and overall survival (HR 3.0, 95%CI 1.3–7.1, *p* = 0.006). Differential expression analysis of 770 immune-related genes demonstrated an impaired CD8 T-cell response in female patients. Our observed ORR of 53% is similar to that described in current literature with durable responses seen in the majority of patients.

## 1. Introduction

Cutaneous squamous cell carcinoma (CSCC) is the second most common skin cancer amongst Caucasians, accounting for 20–25% of skin cancers. Approximately 3–5% of patients with CSCC develop advanced disease, accounting for roughly 1% of all cancer deaths in Australia [1]. Globally, non-melanomatous skin cancers (including CSCC and basal cell carcinomas) account for 0.6% of all cancer deaths [2]. Advanced CSCC comprises locally advanced and metastatic disease not amenable to surgery or radiotherapy [3,4]. Previous systemic treatments including chemotherapy, epidermal growth factor receptor (EGFR) inhibitors and interferon have demonstrated modest benefits only [5].

The emergence of immune checkpoint inhibitors (ICI), however, has been transformative in the management of advanced CSCC. As CSCCs are highly immunogenic with a high tumour mutational burden, there is strong rationale for the use of ICIs [6]. Several clinical trials have demonstrated marked improvements in clinical outcomes with ICIs. The pivotal phase 1/2 trial by Migden and colleagues demonstrated an objective response rate (ORR) of 47.2% in 59 patients with advanced CSCC treated with the programmed death 1 (PD-1) inhibitor, Cemiplimab [7]. A longer follow-up has shown a two-year overall survival (OS) of 61.8%, with median duration of response of 41.3 months [8]. Similar results were demonstrated in phase 2 studies using Pembrolizumab [9,10,11] and Nivolumab [12].

There is limited evidence regarding the use of ICIs for advanced CSCC in real-world clinical practice. Advanced CSCC often occurs in elderly patients with significant comorbidities or patients with a history of significant immunosuppression who are excluded from clinical trials [13,14]. Moreover, there is no established biomarker to assist in the selection of patients with advanced CSCC for immunotherapy. In the phase 2 trial conducted by Migden and colleagues, responses were observed regardless of programmed death ligand 1 (PD-L1) expression or tumour mutational burden [15]. As immunotherapy is used earlier in the treatment of advanced CSCC, such as the (neo)adjuvant settings, it becomes critical to identify patients most likely to respond.

Here, we describe our experience using immunotherapy as treatment for advanced CSCC in a community setting, with a focus on identifying clinical factors predictive of response to treatment. We then investigate the biological basis of our findings using differential expression analysis of 770 immune-related genes.

## 2. Materials and Methods

### 2.1. Study Design

We performed a retrospective review of patients with advanced CSCC treated within the Illawarra Shoalhaven Local Health District (ISLHD) identified from electronic medical records at ISLHD. All patients with locally advanced and metastatic CSCC who received at least one dose of immunotherapy between January 2019 and September 2022 were included for analysis. Immune checkpoint inhibitors (ICI) used were Cemiplimab, Cosibelimab, Pembrolizumab, and Nivolumab. Access to immunotherapy was via patient access programmes, clinical trials, and the pharmaceutical benefits scheme (PBS).

Clinicopathological variables extracted from patient records include: Age, sex, comorbidities (Charlson Comorbidity Index, CCI), baseline lactate dehydrogenase (LDH) levels, previous treatments, primary site and extent of disease. Objective response rates (ORR) were defined as the proportion of patients achieving a partial response (PR) or complete response (CR) to therapy. Disease control rates (DCR) were defined as the proportion of patients achieving either PR/CR or stable disease (SD). Patients with disease control (CR, PR or SD), and a minimum of 6 months duration of at least stable disease, were defined as responders, whilst patients whose best response was progression, or whose disease progressed within 6 months, were defined as non-responders. Progression-free survival (PFS) was calculated as the time from first cycle of immunotherapy to time to progression or death. Overall survival (OS) was calculated as time from diagnosis to death from any cause.

#### 2.1.1. Sample Collection and RNA Extraction

RNA extraction and data analysis was conducted as previously published by Minaei et al. [16]. Formalin-fixed paraffin-embedded (FFPE) tissue samples were obtained from local pathology services where a sufficient sample was available (*n* = 42). Hematoxylin- and Eosin-stained slides were used by an experienced pathologist (SN) to identify areas of the sample with high tumour and low stromal content. The selected areas were extracted from the FFPE tissues using a Quick-Ray^®^ Manual Tissue Microarrayer (UNITMA, South Korea) in 2–3 mm cores.

FFPE cores were de-paraffinized using xylene, and nucleic acids were extracted using an AllPrep DNA/RNA FFPE Kit (80234, Qiagen Hilden, Germany) according to the manufacturer’s instructions. As an initial QC, a NanoDrop^®^ ND-1000 UV-Vis Spectrophotometer (Thermo Fisher Scientific, Waltham, MA, USA) was used to measure A260/280 and A260/230 ratios. Samples passing initial QC were tested for their integrity using a Qubit RNA IQ Assay (Invitrogen, Waltham, MA, USA by Thermo Fisher Scientific) and their gene expression analysed. A total of 28 samples could not be obtained, successfully extracted or failed RNA QC, with 23 samples available for gene expression analysis.

#### 2.1.2. Gene Expression Assay, Data Normalization and Data Analysis

Gene expression of 770 immune-related genes included in the nCounter Human PanCancer IO 360 panel was determined using an nCounter Sprint Profiler (NanoString Technologies, Seattle, WA, USA) as per the manufacturer’s instructions. The amount of purified RNA (150–500 ng) loaded was adjusted according to its quality and integrity.

Technical QC was conducted using NanoNormIter R package as previously published [16,17]. Patient 17 failed QC due to low geometric mean of overall gene expression of the sample and was excluded. No housekeeping gene (*n* = 20) showed significant association with the phenotype of interest as determined using the glm.nb (Negative Binomial Generalized Linear Model) function [18]. Normalization was performed with RUVg using housekeeping genes. After monitoring various principal component analysis (PCA) and relative logarithmic expression (RLE) plots, a parameter k = 3 was considered as appropriate for optimization [19]. At this stage, no further samples were excluded.

Normalized gene expression for all samples can be found in Appendix A. Differential gene expression analyses were performed using DESeq2 [20]. Cook’s outliers were replaced with the predicted geometric mean of the overall expression of the gene across samples (as recommended by DESeq2 documentation). *p*-values were adjusted for multiple-testing using the Benjamini–Hochberg method [21]. Genes were considered differentially expressed between the groups with adjusted *p*-values (p_adj_) < 0.05 and log_2_fold changes |LFC| > 1. The data for all cohort comparisons (including LFC and P_adj_) is included as Appendix A.

Differentially expressed genes (DEGs) were passed into the Enrichr web portal with the panel genes as background to conduct overrepresentation analysis [22,23,24]. Human KEGG (2021) and WikiPathways (2021) were consulted to identify significantly overrepresented pathways [25,26,27,28]. The resulting pathways were visualized using Cytoscape in association with the WikiPathways plugin [29]. To determine immune cell proportions, CibersortX Digital Cytometry was run for 1000 iterations in association with their LM22 signature matrix [30]. Statistical analysis (Student *T*-test) of mean imputed T-cell fractions between male and female patients and visualization of the results were conducted with GraphPad Prism 9 (GraphPad Software, Boston, MA, USA).

### 2.2. Statistical Analysis

Except for the gene expression analysis, all statistical analysis was performed using SAS 9.2 software (SAS Institute, Inc., Cary, NC, USA). To test the association between best response and our collected variables, we performed a chi-squared test for categorical variables and ANOVA for continuous variables. Patient characteristics were compared with chi-squared test. Survival analysis was performed using the Kaplan–Meier method with log-rank test. Univariate Cox proportional hazard regression analyses were used to evaluate the impact of key factors on survival outcomes and to calculate corresponding hazard ratios (HR) and 95% confidence intervals (CIs). All variables significant in the univariate analysis (*p* < 0.05) were included in the multivariate model. All *p*-values < 0.05 (two-sided) were considered statistically significant.

### 2.3. Ethics

The study was approved by the ISLHD Low and Negligible Risk (LNR) Research Review Committee: ISLHD/LNR/2021-111.

## 3. Results

### 3.1. Patient and Tumour Characteristics

A total of 51 patients treated with immunotherapy for advanced/metastatic CSCC were identified; their characteristics are visualised in Table 1. Patients were predominantly male (67%), a feature commonly found across many sites worldwide [31]. The mean age of the cohort was 75 (range 46–93 years). Previous treatment consisted of surgical excision in 38 (74.5%) patients, radiotherapy in 29 (56.9%) patients and chemotherapy in three (5.9%) patients. There were six (12%) patients with a history of significant immunosuppression including myelofibrosis (two patients), chronic lymphocytic leukaemia and hypogammaglobulinaemia (two patients), renal transplant (one patient) and rheumatoid arthritis on immunomodulatory agents (one patient). Out of the six patients with a history of significant immunosuppression, four patients were responders.

The most common primary site of disease was the head and neck (57%). Amongst the entire patient cohort, 23 (45%) patients had locally advanced disease and 28 (55%) patients had metastatic disease; 35% of those with metastatic disease had visceral metastases.

### 3.2. Treatment Outcomes

All patients received anti-PD-1 or PD-L1 therapy with 21 (41%) patients treated with Cemiplimab, whilst 30 (59%) patients received other ICIs including Nivolumab, Pembrolizumab, or Cosibelimab. The median number of cycles of immunotherapy received was six cycles (IQR 3-14). The ORR for all patients was 53%, with six (12%) patients achieving complete responses, and 21 (41%) patients achieving partial responses. In addition, seven (14%) patients had stable disease (for at least 6 months) for a DCR of 67% (these patients were deemed responders), whilst 17 (33%) patients experienced disease progression (these patients were deemed non-responders). The median PFS for the cohort was 39.5 months, and median OS has not been reached with a median follow up of 32 months (range 3.8–82.0 months).

### 3.3. Association between Baseline Clinical Factors and Treatment Efficacy

We analysed the possible association between DCR and baseline clinical factors (Table 1), and PFS and OS (Table 2). In our cohort, sex was the only factor significantly associated with DCR, PFS and OS. The DCR was 85% in males and 41% in females (*p* < 0.0001), with shorter PFS also seen in female patients (median PFS 3.6 months in females versus not reached in males, univariate HR 4.6, 95% CI 1.9–10.8, *p* = 0.0007, Figure 1). There were no differences seen in response between age, site of primary CSCC (head and neck vs other), disease extent, history of significant immunosuppression, receipt of prior surgery or radiotherapy, LDH levels, treatment with antibiotics within a month of commencing immunotherapy, or choice of immunotherapy agent. As sex was the only significant factor in univariate analyses for PFS, no multivariate model was undertaken.

In addition, sex was also a significant predictor of OS in univariate analyses (median OS 13.6 months in females vs not reached in male patients, univariate HR 3.0, 95% CI 1.3–7.1, *p* = 0.006, Figure 2). Although age and comorbidities (CCI) were not associated with response rate or PFS, both were significantly associated with OS in univariate analyses, with older age and high CCI associated with shorter OS (Table 2). After adjusting for age and CCI in multivariate analysis, sex remained a significant independent predictor of OS (multivariate HR 2.5, 95% CI 1.1–6.1, *p* = 0.03, Appendix A). Sex was not significantly associated with other factors including age or CCI, although males were more likely to have visceral metastases (*p* = 0.04, Appendix A), and females were more likely to previously been treated with surgery for advanced disease (*p* = 0.02, Appendix A).

### 3.4. Sex-Based Differential Gene Expression of Immune-Related Genes

We determined the expression of immune-related genes in a subset (based on tissue availability and QC measures) of our study cohort (six female, 17 male). A sex-based comparison yielded 68 significantly differentially expressed genes (p_adj_ < 0.05, |LFC| > 1) (Appendix A). Hierarchical clustering based on the top 20 DEGs show two distinct clusters that reflect the patient sex (Figure 3). In female patients, the expression of genes related to effector T-Cells (e.g., *GZMB*, *CD96*, *CD7*, *CD3G*, *SH2D1A*) was consistently low compared to that in males (Figure 3, Appendix A). This suggests a reduced presence of T-cells within the cored area of CSCC samples from female patients.

Pathway over-representation analysis further corroborates these findings. WikiPathway Analysis yielded a significant enrichment of the DEGs in the “Modulators of TCR signaling and T cell activation (WP5072)” pathway (*p* < 0.05) (Figure 4). Finally, CibersortX-based analysis of immune cell proportions predicts a significant decrease in CD8 T-cells in female patients compared to male patients (*p* < 0.05) (Figure 4). Overall, these results point towards an altered T-cell response and T-cell exclusion from the tumours in female patients, which is consistent with the worse responses to ICIs.

## 4. Discussion

Immunotherapy offers a systemic treatment option with the potential for durable responses and an acceptable toxicity profile. Our purpose was to describe our experience with immunotherapy in a real-world cohort of CSCC patients in Australia, to help inform clinical decision making and identify potential patterns of responses.

Our study found that sex was a significant predictor of response, with male patients having significantly improved DCR, PFS and OS compared to female patients. Of particular note, the association with sex and OS remained significant after adjusting for age and CCI (multivariate HR 2.5, 95%CI 1.1–6.1, *p* = 0.03). Whilst acknowledging the limitations of both the modest sample size and retrospective design of this study, we were unable to find any significant associations with other factors to explain this finding. An observational study by Jang and colleagues [33] also highlighted a difference in response to immunotherapy between males and females. They found that the risk of mortality was 2.06 times higher in female compared with male patients who received combination Ipilimumab and Nivolumab (HR 2.06, 95% CI 1.28–3.32, *p* = 0.003). Ma and colleagues suggest that the influence of sex hormones, genetic differences and overlapping epigenetic alterations underpin the disparities in response to immunotherapy between the male and female sexes [34]. While sex is known to have an effect on the immune response to infections and autoimmune disease, the impact of sex on response to immunotherapy remains unclear, with conflicting results seen in meta-analyses of multiple solid tumours [35]. While there have been significant attempts to explore molecular profiles to better understand the effects of sex on immunotherapy efficacy, ultimately, how sex affects one’s tumour microenvironment and response to immunotherapy remains largely unknown and warrants further exploration [36,37].

Further investigation into the biological mechanism revealed a potentially altered CD8 T-cell response. A study by Budden et al. [38] reported increased aggressiveness of CSCC and increased metastatic potential in CSCC in males compared to females. Validation of these clinical findings in murine models revealed increased anti-tumour immunity as well as increased CD4 and CD8 T-cell infiltration. While this generally leads to milder CSCC in women [38], the cases included in our study present with advanced disease and exclusion of T-cell infiltrates could have occurred during progression of the lesions. However, due to the low numbers of samples and lack of histopathological confirmation of these findings, further investigation in a validation cohort is required to support this hypothesis.

We did not find any other predictors of response to treatment, including factors identified by other series. Two series have found significantly improved response rates in patients with CSCCs arising from the head and neck compared to other sites [39,40]. The phase II study by Munhoz and colleagues reported worse outcomes for patients with prior exposure to radiotherapy [12]. Importantly, we did not find a significant impact of age on the effect of immunotherapy, similar to previous work [41]. It appears that the efficacy of ICIs in advanced CSCC is not dependent or affected by increasing age. Furthermore, a meta-analysis by Kasherman and colleagues demonstrated that ICIs are associated with significant OS improvement compared with control therapies regardless of patient age based on dichotomisation at 65 years old [42].

A review of the literature highlighted a limited number of phase II trials of patients with advanced CSCC treated with immunotherapy (Table 3). The final analysis from the EMPOWER-CSCC-1 phase II study reported an ORR of 50.8% in group 1 (metastatic CSCC treated with Cemiplimab 3 mg/kg 2-weekly), 44.9% in group 2 (locally advanced SCC treated with Cemiplimab 3 mg/kg 2-weekly) and 46.4% in group 3 (metastatic CSCC treated with Cemiplimab 350 mg q3weekly) [8]. KEYNOTE-629 reported an ORR of 50% in locally advanced CSCC and 35.2% in metastatic CSCC treated with Pembrolizumab [11]. The phase II trial by Munhoz and colleagues reported an ORR of 58.3% in patients with locally advanced CSCC treated with Nivolumab [12]. Unfortunately, the impact of sex differences has not been addressed by these studies.

In our study, we found an ORR of 53% with immunotherapy in our cohort of patients with advanced CSCC. This is consistent with the ORRs reported from the phase II clinical trials described above (34.3–58%, Table 3). Similarly, we noted a partial and complete response rate of 41% and 12%, respectively, which mirrors the phase II results. This is despite our cohort being older (47% of patients 79 years and older) and more comorbid (59% of patients with CCI > 5) than the usual clinical trial population. In addition, 12% of patients (*n* = 6) had significant immunosuppression which would have excluded them from the phase II trials, of which 67% were responders. Similar ORRs have been demonstrated in other retrospective studies (31.5–68.0%, Table 3). Overall, this supports the widespread applicability of immunotherapy in the routine care of patients with advanced CSCC.

Despite the impressive responses seen, it is important to note that approximately half of CSCC patients do not respond to immunotherapy, and it remains a key challenge to identify patients who respond best, to maximise the therapeutic benefit. Moreover, as immunotherapy is used earlier in patient treatment, including in the neoadjuvant settings, it is critical to identify patients most likely to respond so as to not delay curative intent treatment. While we do not suggest that female sex is a reason to withhold immunotherapy, these provocative data highlight the need for further translational studies to unravel the molecular and immunological drivers of this observation. There are no currently accepted predictive biomarkers for immunotherapy in CSCC. Similar to other solid malignancies, responses are seen in patients regardless of PD-L1 status [8,11]. Some small series have suggested tumour mutational burden as a possible biomarker, although this feature is generally high in CSCC [43].

The limitations of our study include its smaller sample size. While there is increasing recognition of the true incidence of advanced CSCC, there is a paucity of literature reporting clinical trials and real-world outcomes. Our study is also limited by its retrospective design, and the results, whilst intriguing, warrant prospective validation.

## 5. Conclusions

In conclusion, our study represents a real-world experience of using immunotherapy in an Australian cohort of patients with advanced CSCC and supports current data that immunotherapy is an effective treatment with durable responses. Whilst half of patients diagnosed with advanced CSCC will respond to immunotherapy, there remains an urgent clinical need to identify biomarkers to predict response. Our study has found sex to be a significant predictor of response, with male patients responding better to immunotherapy when compared to female patients. Furthermore, our differential expression analysis of 770 immune-related genes suggests an impaired CD8 T-cell response in female patients. However, we highlight the retrospective nature of our study and potential for significant biases. Our findings highlight the need for more prospective studies to investigate biomarkers of response, and the potential relationship between sex and response to immune checkpoint inhibition.

## Figures and Tables

**Figure 1 cancers-15-05026-f001:**
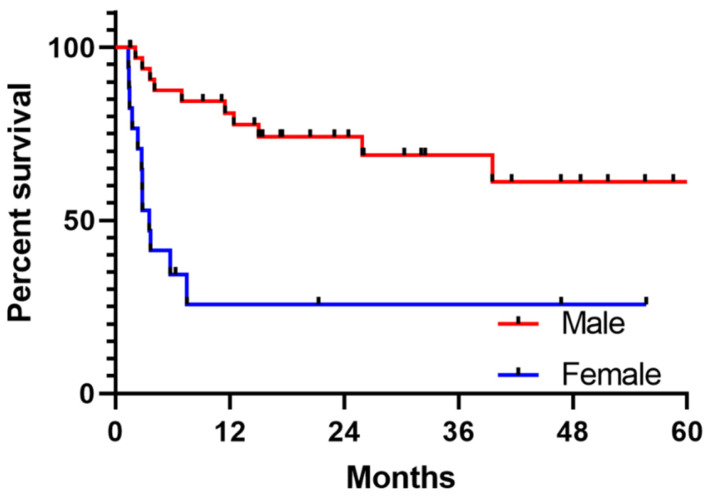
Kaplan–Meier curve for the probability of PFS between male and female patients. The median PFS was 3.6 months in female patients versus not reached in male patients, HR 4.6, 95% CI 1.9–10.8, *p* = 0.0007.

**Figure 2 cancers-15-05026-f002:**
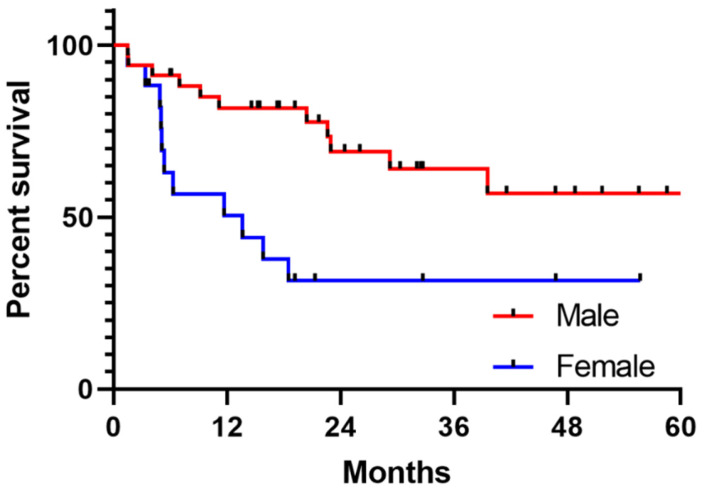
Kaplan–Meier curve for OS between male and female patients. The median OS was 13.6 months in female patients versus not reached in male patients, HR 3.0, 95% CI 1.3–7.1, *p* = 0.006.

**Figure 3 cancers-15-05026-f003:**
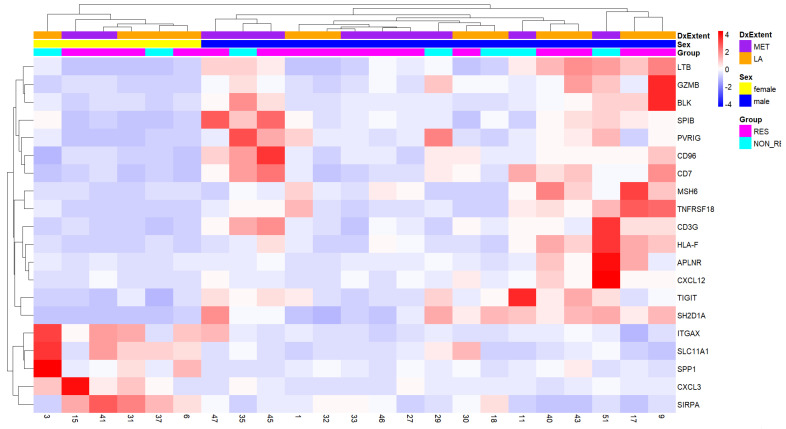
Hierarchically clustered heatmap of top 20 DEGs based on sex in advanced CSCC. Differential expression analysis of the normalized gene counts was conducted and the top 20 DEGs (p_adj_ < 0.05, |LFC| > 1) were plotted using pheatmap [32]. Expression values are scaled across rows for comparability between patients but not genes. Each column represents a patient with identifier number listed below each column. Male and female patients form two distinct clusters. DxExtent—Disease Extent, RES—responder, NON_RES—non-responder, MET—metastatic, LA—locally advanced.

**Figure 4 cancers-15-05026-f004:**
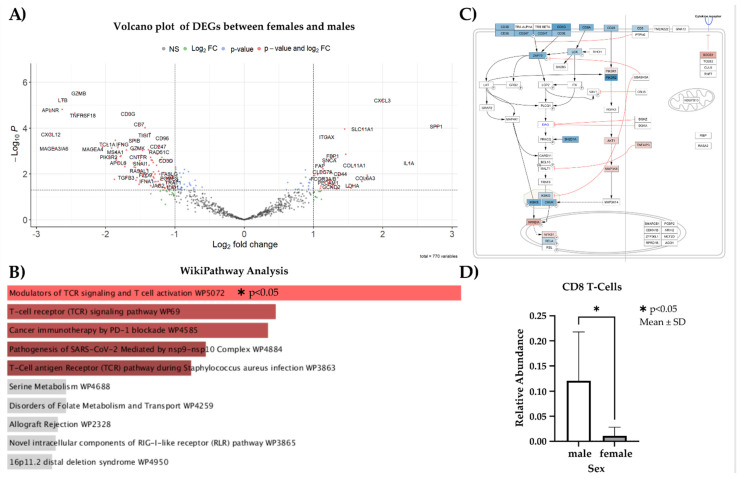
Differential gene expression analysis between male and female patients with advanced CSCC. (**A**) Volcano plot of DEGs. Cut-off values for adjusted *p*-values and |LFCs| are 0.05 and 1, respectively. (**B**) Pathway over-representation analysis of DEGs. Analysis and visualisation conducted using the Enrichr webportal. (**C**) Visualisation of the Modulators of TCR signalling and T-cell activation pathway as per WikiPathways. Warmer colours indicate higher expression, and cooler colours indicate reduced expression in female patients. (**D**) Mean CD8 T-cell proportions in female and male patients as predicted by CybersortX.

**Table 1 cancers-15-05026-t001:** Analysis of patients with advanced cutaneous squamous cell carcinoma treated with immunotherapy.

Variables	Total *n* = 51	Responders (CR/PR/SD) *n* = 34 (67%)	Non-Responders *n* = 17 (33%)	*p* Value (Chi Sq)
**Sex:**				
Male	34	29 (85)	5 (15)	<0.0001
Female	17	7 (41)	10 (58)	
**Age:**				
<69	15	11 (73)	4 (27)	0.78
69–79	12	8 (66)	9 (33)	
≥79	24	15 (63)	8 (37)	
**CCI**				
CCI < 5	21	15 (71)	6 (29)	0.54
CCI ≥ 5	30	19 (63)	11 (37)	
**Site of Primary CSCC**				
Head and neck	29	20 (69)	9 (31)	0.77
Other	22	14 (64)	8 (36)	
**Disease extent**				
Locally advanced	23	15 (65)	8 (35)	0.84
Metastatic	28	19 (69)	9 (32)	
**Presence of visceral metastases**	18	14 (75)	4 (22)	0.13
**Significant Immunosuppression**				
No	45	30 (66)	15 (34)	0.95
Yes	6	4 (67)	2 (33)	
**Antibiotics Prior to Starting ICI**				
No	43	28 (65)	15 (35)	0.58
Yes	8	6 (75)	2 (25)	
**Previous radiotherapy**				
No	22	13 (59)	9 (41)	0.32
Yes	29	21 (72)	8 (28)	
**Previous surgery for advanced disease**				
No	14	8 (57)	6 (43)	0.37
Yes	37	26 (70)	11 (30)	
**Elevated LDH**	9	8 (88)	1 (12)	0.18
**Immunotherapy agent**				
Cemiplimab	21	14 (66)	7 (33)	0.99
Other agents	30	20 (66)	10 (33)	

CCI—Charlston comorbidity index; LDH—Lactate dehydrogenase; Other agents—Pembrolizumab, Cosibelimab, Nivolumab.

**Table 2 cancers-15-05026-t002:** Overall survival and progression-free survival.

Variables	Overall Survival		Progression-Free Survival	
	HR (95% CI)	*p*	HR (95% CI)	*p*
**Age**				
<69	1		1	
69–78	4.0 (0.8–21.1)	0.007	1.4 (0.4–4.8)	0.40
≥79	6.9 (1.6–30.7)		2.0 (0.7–5.8)	
**Sex**				
Male	1		1	
Female	3.0 (1.3–7.1)	0.006	4.6 (1.9–10.8)	0.0007
**CCI**				
<5	1		1	
5 or more	3.1 (1.1–8.4)	0.02	1.4 (0.6–3.3)	0.46
**Site of Primary SCC**				
Head and neck	1		1	
Other	1.0 (0.4–2.5)	0.91	1.2 (0.5–2.7)	0.60
**Disease extent**				
Locally Advanced	1		1	
Metastatic	1.0 (0.4–2.4)	0.98	0.99 (0.4–2.9)	0.97
**Visceral Mets**				
No	1		1	
Yes	0.7 (0.2–2.2)	0.57	0.47 (0.2–1.5)	0.19
**Significant Immunosuppression**				
No	1		1	
Yes	1.7 (0.6–5.2)	0.30	2.3 (0.9–6.3)	0.08
**Previous Radiotherapy**				
No	1		1	
Yes	0.9 (0.4–2.2)	0.85	0.9 (0.3–2.0)	0.76
**Previous Surgery for Advanced Disease**				
No	1		1	
Yes	0.7 (0.3–1.6)	0.37	0.8 (0.3–1.9)	0.55
**Antibiotics prior to Immunotherapy**				
No	1		1	
Yes	0.53 (0.1–2.3)	0.39	0.5 (0.1–2.2)	0.34
**Elevated LDH**				
No	1		1	
yes	1.1 (0.4–3.2)	0.73	0.8 (0.3–2.3)	0.67
**Immunotherapy Agent**				
Cemiplimab	1		1	
Other agents	2.1 (0.8–5.4)	0.10	0.5 (0.1–2.2)	0.88

CCI—Charlston comorbidity index; LDH—Lactate dehydrogenase; Other agents—Pembrolizumab, Cosibelimab, Nivolumab.

**Table 3 cancers-15-05026-t003:** Literature review on immunotherapy clinical trials in advanced CSCC.

Trial	Agent		Patient Population	Total ORR
Prospective Trials				
**Maubec 2020 [9], Hughes 2021 [11]**	Pembrolizumab	Phase II	159 patients (54 LA, 105 recurrent/metastatic)	50.0% (27/54) LA 35.2% (37/105) recurrent/metastatic
**Migden 2022 [8]**	Cemiplimab	Phase II	59 patients LA and metastatic (Group 1) 78 patients LA 56 patients metastatic	50.8% (30/59) 44.9% (35/78) 46.4% (26/56)
**Munhoz 2022 [12]**	Nivolumab	Phase II	24 LA	58.3% (14/24)
**Retrospective Trials**				
**In 2020**	ICI	Retrospective	26 LA and metastatic	42.3%
**Hanna 2020**	ICI	Retrospective	61 LA and metastatic	31.5%
**Salzmann 2020**	ICI	Retrospective	46 LA and metastatic	58.7%
**Baggi 2021**	Cemiplimab	Retrospective	131 LA and metastatic	58.0%
**Samaran 2022**	ICI	Retrospective	63 LA and metastatic	57.1%
**Hasmat 2023**	Cemiplimab	Retrospective	19 LA and metastatic	68.0%

ICI = Immune checkpoint inhibitor; LA = Locally advanced disease.

## Data Availability

Data are available at reasonable request through the corresponding author.

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
