# Peer review of "Sex as a Predictor of Response to Immunotherapy in Advanced Cutaneous Squamous Cell Carcinoma"

_cancers, 2023, doi:10.3390/cancers15205026_

Round 1

Reviewer 1 Report

Article (cancers-2642965): Sex as a predictor of response to immunotherapy in advanced cutaneous squamous cell carcinoma

Authors have done a retrospective study on the cohort of cutaneous squamous cell carcinoma (CSCC) patients in Australia treated with immunotherapy.  Using different systems biology tools (gene expression analyses on 770 immune-related genes, high-throughput dissection of cellular heterogeneity in obtained genomic profiles, computational biological pathways visualization, etc.) they have produce interesting piece of work, revealing that male patients were more likely to have better responses to particular  immunotherapy in comparison to female patients. Furthermore, this gender linkage was observed for the overall survival, again in favour of males.

This is a nice study, but requests some further clarifications.  I have suggested minor corrections, which, I hope, could be met and arranged by the authors. I hope this will improve the manuscript, thus it could be accepted for a publication.

The authors need to correct (or present a better explanation on) the following: 

1.      In the Introduction part, authors provide some statistical details on spread of the disease in Australia, but it would be nice to give some info on the worldwide situation (lines 53-54).

2.      It should be separated Pvalues, line 134.

3.     Missing explanation for MET and LA abbreviations in Figure 3. legend (suppose Metastatic and Locally advanced, and have to be added in line 245).

4.   It would be nice to perform at least one functional assay on CD8 T-cell counting/sorting and to compare with CybersortX obtained results (Figure 4. D).

5.      Please standardize the writing of drugs (with or without a capital letter), since it is changing throughout the text (for instance: lines 64, 66, 182, 308,309,311, 313)

6.      Table 3. In the last row is written metastastic, please correct.

7.  Any thoughts or reflections on: “12% of patients (n=6) had significant immunosuppression which would have excluded them from the phase II trials, of which 67% were responders.….. “ Please, comment/hypothesise what might contribute to the fact that immunosuppressed patients still were able to respond desirably. Which cells were targeted by particular immunosuppressive drugs, and what sorts of cells/mechanisms might be involved in the response to CSCC immunotherapy?

8.      Acknowledgments: None.

-         It would be nice to thank all the patients who were included in the study (what is voluntary act), since without data derived from their treatments and tissue samples they would not have the study!

Reviewer 2 Report

This manuscript presents results from a retrospective study of the use of anti-PD1 immunotherapy for advanced cutaneous squamous cell carcinoma in a real world cohort of patients. It reports the interesting finding that females were less likely to respond to the therapy than males, even after controlling for age, co-morbid conditions, prior treatments and stage of disease. This result was further strengthened by gene expression analysis of preserved tissue samples that showed a significant difference in gene expression T-cell specific genes in tumor tissue from females compared to males.  The presentation is clear and complete, with adequate discussion of limitations. Figure 4c is especially helpful. I have no suggestions for improvement. This report seems likely to initiate a re-examination of previous ICI trials for sex differences, and prospective studies to try to improve ICI efficacy in females.

Author Response

Dear reviewer 2, 

Thank you very much for your feedback. No changes were made from this end as there were no suggestions for changes. 

Many thanks and kind regards,

Nicholas Yeo